# HSPA1L Enhances Cancer Stem Cell-Like Properties by Activating IGF1Rβ and Regulating β-Catenin Transcription

**DOI:** 10.3390/ijms21186957

**Published:** 2020-09-22

**Authors:** Soo-Im Choi, Jei-Ha Lee, Rae-Kwon Kim, Uhee Jung, Yeon-Jee Kahm, Eun-Wie Cho, In-Gyu Kim

**Affiliations:** 1Department of Radiation Biology, Environmental Radiation Research Group, Korea Atomic Energy Research Institute, Daejeon 34057, Korea; sichoi@kaeri.re.kr (S.-I.C.); jeiha7@gmail.com (J.-H.L.); rkim@kaeri.re.kr (R.-K.K.); uhjung@kaeri.re.kr (U.J.); kahmyj@kaeri.re.kr (Y.-J.K.); 2Department of Radiation Biotechnology and Applied Radioisotope, Korea University of Science and Technology, Daejeon 34057, Korea; 3Rare Disease Research Center, Korea Research Institute of Bioscience and Biotechnology, Daejeon 34141, Korea; ewcho@kribb.re.kr

**Keywords:** HSPA1L, ALDH1, cancer stem cell, EMT, β-catenin

## Abstract

Studies have shown that cancer stem cells (CSCs) are involved in resistance and metastasis of cancer; thus, therapies targeting CSCs have been proposed. Here, we report that heat shock 70-kDa protein 1-like (HSPA1L) is partly involved in enhancing epithelial–mesenchymal transition (EMT) and CSC-like properties in non-small cell lung cancer (NSCLC) cells. Aldehyde dehydrogenase 1 (ALDH1) is considered a CSC marker in some lung cancers. Here, we analyzed transcriptional changes in genes between ALDH1high and ALDH1low cells sorted from A549 NSCLC cells and found that HSPA1L was highly expressed in ALDH1high cells. HSPA1L played two important roles in enhancing CSC-like properties. First, HSPA1L interacts directly with IGF1Rβ and integrin αV to form a triple complex that is involved in IGF1Rβ activation. HSPA1L/integrin αV complex-associated IGF1Rβ activation intensified the EMT-associated cancer stemness and γ-radiation resistance through its downstream AKT/NF-κB or AKT/GSK3β/β-catenin activation pathway. Secondly, HSPA1L was also present in the nucleus and could bind directly to the promoter region of β-catenin to function as a transcription activator of β-catenin, an important signaling protein characterizing CSCs by regulating ALDH1 expression. HSPA1L may be a novel potential target for cancer treatment because it both enhances IGF1Rβ activation and regulates γβ-catenin transcription, accumulating CSC-like properties.

## 1. Introduction

Cancer stem cells (CSCs), first revealed in acute myeloid leukemia, exist in a variety of solid tumors, including lung cancer [1,2,3]. Because CSC is highly resistant to conventional therapy, it is difficult to eradicate it with ionizing radiation or anticancer drugs [4,5]. Therefore, CSC is being studied by many research groups as it is an important target for cancer treatment [6,7]. Among cancers, lung cancer has a high mortality rate, so new therapeutic strategies are required [8]. In particular, non-small cell lung cancer (NSCLC) is difficult to treat because it is resistant to anticancer agents, such as ionizing radiation and chemical-targeted drugs. Therefore, patients with NSCLC recur frequently [9]. A recent studies have proposed that small CSC subpopulations mediate the selection of radiation-resistant NSCLC cells [10]. CSC-like cells are commonly characterized and isolated by specific CSC surface markers, including transmembrane glycoproteins, CD44, CD133, and CD326 (EpCAM) in solid tumors [11,12,13]. Like cell surface glycoproteins, aldehyde dehydrogenase 1 (ALDH1), which is involved in self-renewal, differentiation and detoxification, is overexpressed in cells from lung cancer patients with poor prognoses [14,15]. Therefore, ALDH1 is also a critical CSC marker in various cancers, including lung cancer [16]. In this study, we sought to find novel factors and associated signaling pathways involved in CSCs characterization.

Heat shock proteins (HSPs) are produced in response to various stresses or diseases, such as hypoxia, ischemia, metabolic disorders and inflammation in eukaryotic cells [17]. In addition to functioning as molecular chaperones in cells [18], circulating HSPs play various physiological roles by regulating autoimmune responses, T-cell responses and cytoprotection [19,20,21]. HSPs are classified by their molecular weight. The heat shock protein 70 family (HSPA) is divided into seven evolutionarily distinguished groups. HSPA is a molecular chaperone that guides the protein-folding, translocation and degradation pathways. HSPA has also been associated with various diseases and pathophysiologies, including ischemic injury, neurodegenerative diseases, and cancer [22]. Among the HSPA isotypes, heat shock 70-kDa protein 1-like (HSPA1L) is included in group VI along with HSPA1A and HSPA1B, the main members of HSPA [23]. HSPA1L is highly and constitutively expressed in male germ lines and protects male germ cells from external stresses such as heat and oxidative stress [24,25]. Therefore, HSPA1L gene polymorphisms in the region encoding the client-binding domain are reported to be associated with diseases such as male infertility [26,27]. However, its detailed functions are unclear. Moreover, despite reports that genetic variations of HSPA1L may be linked to cancer [28,29], the pathways and mechanisms by which HSPA1L is involved in cancer stemness and poor prognoses are unclear. Here, we showed that upregulated HSPA1L in ALDH1high cells sorted from NSCLC A549 cells was involved in insulin-like growth factor-1 receptor β IGF1Rβ activation and enhanced the CSC-like properties, resistance, and malignancy via either the AKT/NF-κB or AKT/GSK3β/β-catenin signaling pathways. In addition, HSPA1L was directly involved in the transcription of β-catenin, a transcription regulator of ALDH1 expression.

## 2. Results

### 2.1. HSPA1L Was Upregulated in ALDH1high Cells Isolated from A549 Cells

Determining factors that characterize CSCs is important in overcoming cancer resistance and relapse. Despite accumulating evidence that ALDH1 plays an important role in CSC properties, studies related to specific mechanisms and pathways involved in regulating ALDH1 expression are still insufficient. To select the factors expected to be involved in regulating ALDH1 expression, ALDH1high and ALDH1low cells were sorted from A549 cells using flow cytometry based on ALDEFLUOR analysis. Sphere-forming assays showed that ALDH1high cells had higher self-renewal capacity for CSCs than did ALDH1low cells (Figure 1A). cDNA microarray analysis was used to pinpoint genes associated with CSC characteristics between the ALDH1high and ALDH1low cells, revealing 4000 differentially expressed human genes (data not shown). Among these genes, HSPA1L expression was approximately 35 times greater in ALDH1high cells than in ALDH1low cells. Other HSPA isotypes, such as HSP70B′ and HSPA8, also showed large differences in gene expression, among which HSPA1L had the highest expression difference (Figure 1B, Appendix A). Based on the DNA chip analysis, we confirmed that using Western blot assays, the protein HSPA1L levels were more highly upregulated in ALDH1high cells than in ALDH1low cells. Additionally, cellular levels of the transcription regulators, Sox2, Oct4, and Nanog, which are involved in cell stemness, were more upregulated in ALDH1high cells than in ALDH1low cells. In particular, β-catenin involved in transcription of ALDH expression was increased in ALDH1high cells. Cellular levels of the CSC surface marker proteins CD133 and CD44 were also increased (Figure 1C).

### 2.2. HSPA1L Promoted Self-Renewal and Tumorigenic Capacity in Lung Cancer Cells

Although many HSP functions have been identified, little is known about the function of the HSPA1L in cancer cells. Therefore, in this study, to investigate whether HSPA1L was involved in the enrichment of stem cells in lung cancer cells, A549 cells, an adenocarcinoma cell line with a high radiation resistance and a high cellular level of ALDH1, and H460 cells with a relatively low radiation resistance and low cellular level of ALDH1 were used. A549 and H460 cells were cultured in serum-free medium containing epidermal growth factor (EGF) and basic fibroblast growth factor (bFGF) to produce spheroids. Single-cell analysis revealed that suppressing HSPA1L expression markedly delayed spheroid formation. The size of the spheroids was significantly decreased. Conversely, forcibly overexpressing HSPA1L led to aggressive and rapid spheroid formation (Figure 2A). A soft agar assay showed that HSPA1L regulation affected the number of colonies. Forced inhibition of HSPA1L expression using siRNA reduced the number of colonies, whereas overexpression of HSPA1L increased the number of colonies (Figure 2B). CSCs mediate tumor resistance to ionizing radiation and relapse [10]. Thus, controlling genes involved in CSC properties enables reducing tumor resistance to ionizing radiation and maximizes treatment efficiency. One aim of this study was to determine whether HSPA1L was involved in tumor resistance to ionizing radiation, a CSC characteristic. To test this hypothesis, we first examined whether HSPA1L was required for clonal formation in A549 and H460 cells using anchorage dependence. Consequently, colony formation was suppressed in the group with the reduced HSPA1L expression. In addition, exposing A549 and H460 cells with suppressed HSPA1L expression to ionizing radiation significantly increased the cells’ sensitivity to ionizing radiation compared with that of the control group. Conversely, the number of colonies was increased in cells overexpressing HSPA1L compared with that of the control group. Exposing HSPA1L-overexpressing cells to ionizing radiation increased the resistance to ionizing radiation (Figure 2C). These results suggest that HSPA1L is involved in cell proliferation, self-renewal ability, and radiation resistance in lung cancer cells. To confirm this result, Western blot analysis was performed to investigate changes in the typical CSC-characterizing markers, CD44, ALDH1A1 and ALDH1A3, as well as the CSC-related transcription factors, Sox2, Oct4, Nanog, and β-catenin. Cellular CSC marker protein levels were decreased in the HSPA1L-suppressed lung cancer cells but increased in cells overexpressing HSPA1L (Figure 2D). Immunocytochemical analysis confirmed that cellular ALDH1A1 and CD44, representative CSC-characterizing biomarkers, significantly decreased with suppression of HSPA1L expression (Figure 2E).

### 2.3. HSPA1L Promoted Migratory and Invasive Properties in Lung Cancer Cells via Epithelial-Mesenchymal Transition (EMT)

A recent study reported a direct link between EMT progression and acquisition of CSC properties [30]. Therefore, EMT and CSCs have similar signaling pathways and mechanisms. The most distinctive feature of EMT is associated with cancer metastasis owing to increased migration and invasion abilities of cancer cells. Effectively controlling the signaling pathways or mechanisms involved in EMT can help prevent cancer metastasis. However, whether HSPA1L is related to EMT is unclear. We observed whether HSPA1L affected the EMT process in A549 and H460 cells. Inhibiting HSPA1L expression with siRNA in A549 and H460 cells significantly reduced the cancer cell mobility and invasiveness. Conversely, forcibly overexpressing cells using the HSPA1L expression vector increased their mobility and invasiveness. Moreover, inhibiting HSPA1L expression in A549 and H460 cells altered the cell morphology from a spindle shape to a cobble-like shape, whereas HSPA1L-overexpressing cells became spindle shaped (Figure 3A). Wound-healing assays were also used to confirm changes in cell migration capacity via HSPA1L. Suppressing HSPA1L expression via siRNA in A549 and H460 cells significantly decreased the wound-healing capacity compared with that of the control group, whereas overexpressing HSPA1L accelerated the wound-healing capacity (Figure 3B and Appendix A). To confirm these results, we examined the cellular levels of the EMT markers, E-cadherin, N-cadherin, and Vimentin, and the EMT-related transcription factors, Snail and Twist, via Western blotting. Inhibiting HSPA1L expression increased the cellular E-cadherin levels, which is suppressed by Snail, and decreased the cellular N-cadherin, Vimentin, Twist, and Snail levels, thus suppressing metastatic dissemination. In contrast, overexpressing HSPA1L yielded the opposite results (Figure 3C). Immunofluorescence staining confirmed that inhibiting HSPA1L expression upregulated the E-cadherin levels and downregulated the cellular Vimentin levels (Figure 3D).

### 2.4. HSPA1L/Integrin αV Was Involved in the Activation of IGF1Rβ and its Downstream Signals, AKT/NF-κB p65 and AKT/GSK3β/β-Catenin Pathway

IGF1Rβ is highly activated in NSCLC patients with poor prognoses and low survival rates [31]; thus, IGF1R is an attractive therapeutic target for lung cancer [32]. We investigated whether HSPA1L affected IGF1Rβ activation and its downstream signaling activity. An immunoprecipitation assay revealed that HSPA1L directly interacted with IGF1Rβ (Figure 4A), suggesting that the HSPA1L/IGF1Rβ complex regulated IGF1Rβ activation (autophosphorylation). Previous studies have shown that collaboration between IGF1Rβ and integrin αV increases activation of the IGF1Rβ/PI3K/AKT pathway by IGF1 [33]. We showed that HSPA1L also bound directly to integrin αV as well as IGF1Rβ (Figure 4A,B). We also confirmed direct binding between IGF1Rβ and integrin αV (Figure 4C); therefore, inhibiting integrin αV expression by siRNA-inactivated IGF1Rβ (Figure 4D). These results strongly suggest that the complete IGF1Rβ–HSPA1L–integrin-αV triple complex causes HSPA1L to function as intracellular signaling or an activator molecule to activate IGF1Rβ. Confocal microscopy revealed the complex formation of HSPA1L/IGF1Rβ and integrin-αV/IGF1Rβ. A549 cells stained with anti-IGF1Rβ antibody showed dot-shaped signals, which were similar to dot-shaped images in cells stained with anti-HSPA1L and anti-integrin-αV antibody (Figure 4E). IGF1Rβ is a receptor tyrosine kinase that activates the PI3K/AKT/NF-κB signaling pathway, which is closely associated with cell growth and anticancer resistance in NSCLC cells [34,35]. Inhibiting HSPA1L expression via siRNA in A549 and H460 cells inactivated IGF1Rβ/AKT/NF-κB p65 (dephosphorylation) (Figure 4F). Because NFκB is an important downstream transcriptional factor of the PI3K/AKT signaling pathway, its inhibition enhances sensitivity to γ-radiation-induced cell death. Thus, its inhibitors are potential anticancer drug candidates [36]. Overexpressing HSPA1L increased the phosphorylation-mediated activation of the PI3K/AKT signaling pathway. Therefore, HSPA1L is closely related to IGF1Rβ activation and consequently activates its downstream signals to promote cell proliferation and malignant cancer cell progression. HSPA1L activated the IGF1Rβ/AKT/GSK3β pathway (Figure 4F), which stabilized Snail and reinforced the EMT capacity [37]. HSPA1L also partly affected stemness and EMT in cancer cells by regulating ALDH1 expression via β-catenin stabilization by the IGF1Rβ/AKT/GSK3β activation pathway. Therefore, immunocytochemical analysis confirmed that using siRNA to inhibit HSPA1L downregulated the cellular β-catenin level (Figure 4G). Accordingly, flow cytometry analysis using the ALDEFLUOR assay showed that forcing HSPA1L overexpression increased the protein ALDH1 levels, whereas inhibiting HSPA1L expression decreased the cellular ALDH1 level (Figure 4H).

### 2.5. HSPA1L Bound Directly to the Specific Promoter Region of β-Catenin in the Nucleus and Functioned as a Transcription Activator of β-Catenin

HSPA1A, of the main HSPA (HSP70) family, translocates to the nucleus, and some studies have proposed its role [38]. However, it is not even known whether HSPA1L exists in the nucleus of cancer cells. Fractionating cells into the cytoplasm and nucleus revealed that HSPA1L was present in both (Figure 5A). Moreover, the results of immunofluorescence staining in A549 and H460 cell were the same as those of the Western blot (Figure 5B). Thus, HSPA1L has other functions in the cytoplasm and nucleus. β-catenin/TCF is involved in EMT-associated CSCs and therapeutic resistance properties by functioning as an important transcription factor for ALDH1 regulation [39,40]. In this study, along with β-catenin stabilization by the HSPA1L/IGF1Rβ/AKT/GSK3β activation pathway, we identified that HSPA1L also regulated the transcription of β-catenin expression (Figure 5C). Thus, we performed chromatin immunoprecipitation (ChIP) analysis to confirm that HSPA1L was partly involved in obtaining CSC and EMT characteristics by directly activating transcriptional upregulation of β-catenin. Interaction between the DNA-binding histone protein and HSPA1L is essential for histone remodeling and transcriptional activation. HSPA1L antibody was used to investigate whether the HSPA1L protein binds directly to the specific promoter region of β-catenin. To determine the binding site, we examined the sequences that may have important promoter functions, including nucleotides in the region around the exon 1 of β-catenin. Previous studies showed that the β-catenin promoter region contains SP1 (8 sites), AP1 (1 site), AP2 (12 sites), an NF-B binding site (1 site), a UCR core, and an EGR-1 (1 site; Appendix A). Moreover, a TATA box exists at position 29 of the β-catenin 5′-flanking region [41]. We assumed that the HSPA1L protein could bind directly to the β-catenin promoter region, including the TATA box, where a genetic sequence can be read and decoded. Accordingly, based on the nucleotide sequence of the previously identified β-catenin promoter region, we prepared P1 (−103 to +42 nts), P2 (−298 to −139 nts), P3 (−395 to −216 nts), and P4 (0 to +166 nts; Figure 5D and Appendix A). The ChIP assay results confirmed that HSPA1L was directly bound to the promoter P1 site (Figure 5E). In addition, luciferase activity was measured by constructing vectors for P1, P2, P3, and P4. We believe that the P1 site (−103 to +42 nts), including the TATA box, may be a potent promotor region that binds with HSPA1L for β-catenin transcription.

### 2.6. HSPA1L/β-Catenin Axis Was Involved in EMT-Associated CSC Characteristics by Regulating ALDH1 Expression

To confirm whether the HSPA1L/β-catenin axis is involved in EMT-associated CSC characteristics, β-catenin expression was suppressed with siRNA. Inhibiting β-catenin expression slowed A549 cell anchorage-independent cell growth and sphere formation and reduced invasion and migration abilities in lung cancer cells (Figure 6A,B). Therefore, inhibiting β-catenin expression in A549 cells decreased the cellular levels of the representative CSC markers, Sox2 and Oct3/4 (Figure 6C). Cellular levels of the EMT markers, N-cadherin, Vimentin, Twist, and Snail, were also reduced, and the E-cadherin level was increased (Figure 6D). ALDOFLUOR analysis confirmed that β-catenin significantly affected cellular ALDH1 levels, strongly suggesting that the HSPA1L/β-catenin pathway plays an important role in accumulating stemness properties via partial ALDH1 regulation. On the other hand, inhibiting β-catenin did not affect the cellular level of HSPA1L, suggesting that HSP1AL is an upstream regulator of β-catenin (Figure 6E). Thus, HSPA1L enhanced CSC-like properties in cancer cells by both activating IGF1Rβ and regulating β-catenin transcription (Figure 7).

## 3. Discussion

HSPA family members are major stress-inducing proteins that play various roles depending on their location. In addition to being potent neurodegeneration suppressors in schizophrenia and Alzheimer′s disease [42,43], increased intracellular HSPA levels play key roles in recovering from stresses, such as radiotherapy [44]. In contrast, extracellular HSP70s are potent stimulators of the innate immune system and mediators of antitumor immunity [45]. HSPAs are also abundant in cancer, providing selective advantages to malignant cells by suppressing various apoptotic cell death pathways, promoting angiogenesis, enhancing metastasis, and bypassing cellular senescence programming. Despite their diverse roles, one HSPA isotype, HSPA1L, is not a stress-inducing protein. Therefore, research on its function has been limited in several diseases, including cancer. A recent study showed that HSPA1L is involved in accumulating prion proteins in patients with colorectal cancer recurrence [46]. Despite continued reports that HSPA1L polymorphisms are closely associated with various diseases, little is known about the specific signaling or mechanism by which HSPA1L functions in cancer cell stemness. In this study, we confirmed for the first time that HSPA1L was highly expressed in ALDH1high CSC-like cells isolated from an NSCLC cell line and was involved in stemness and γ-radiation resistance of cancer cells by activating IGF1Rβ and regulating β-catenin transcription.

IGF1R, a transmembrane receptor tyrosine kinase upregulated in NSCLC, is correlated with tumor progression and patient prognosis [47]. Binding of IGF1 or IGFII, peptide hormones secreted by many cancer cells and cancer-associated fibroblasts in a autocrine or paracrine manner to extracellular IGF1R domains, enhances expression of biomarkers involved in stemness, such as Nanog, Sox2, and Oct3/4 via cytoplasmic IGF1Rβ activation and IGF1Rβ downstream signaling activation pathways, including PI3K/AKT pathways [34,48,49]. Therefore, the signaling pathways triggered by activated IGF1Rβ are involved in tumor stemness accumulation and malignant cell survival. Integrins are transmembrane heterodimers that form receptors on the cell surface, using extracellular matrix (ECM) components as ligands [50]. Fibronectin, laminin, fibrinogen, and vitronectin are ECM components that act as extracellular ligands of integrins [51], which are involved in signal transduction when ligand binding occurs. Ligation with integrins triggers many signaling events that regulate cell behavior, including proliferation, survival, apoptosis, polarity, motility, gene expression, and differentiation [52]. Integrin receptors are also involved in growth factor signaling. Studies have shown that integrin aVβ3 plays a critical role in IGF1R signaling by binding directly to IGF [53]. Therefore, integrin antagonists can suppress growth factor signaling.

In contrast, our studies have shown that HSPA1L may function as a cytoplasmic regulator for autoactivation of IGF1Rβ. For this purpose, integrin αV appears to support HSPA1L-mediated IGF1Rβ activation by forming the HSPAIL–integrin αV complex. First, we confirmed that HSPA1L binds to both IGF1Rβ and integrin αV in the cytoplasm to regulate the downstream signaling pathways of IGF1Rβ. Therefore, the IGF1Rβ–HSPA1L–integrin αV complex activated the AKT/NF-κB and AKT/ GSK3β/β-catenin pathway. More importantly, in addition to the IGF1Rβ-associated β-catenin stabilization pathway, HSPA1L directly affected β-catenin expression by activating its transcription. Thus, HSPA1L can partially enhance stemness of cancer cell via direct or indirect regulation of the β-catenin/ALDH1 axis.

Deregulation of IGF1R signal transduction inhibits tumor development, progression, and metastasis. Therefore, IGF1R is considered an emerging target for lung cancer treatment, and IGF1R-targeted agents are in advanced stages of clinical development. Nevertheless, these agents have several disadvantages, including acquired resistance and toxic adverse effects. In normal states, IGF1R is involved in glucose and energy metabolism by interacting with adapter proteins, such as insulin receptors and insulin receptor substrates. Therefore, to minimize the influence on the normal physiological functions of IGF1R, the cancer cell-specific downstream targets and signaling pathways of IGF1R must be studied. Therefore, the results herein indicate that HSPA1L is an important potential intracellular activator of IGF1Rβ Accordingly, HSPA1L may be a promising new therapeutic target for treating lung cancer.

## 4. Materials and Methods

### 4.1. Cell Cultures

The A549 and H460 human lung cancer cell lines were obtained from the Korea Cell Line Bank (Seoul, Korea) and grown in RPMI 1640 medium supplemented with 10% (*v*/*v*) fetal bovine serum (FBS; Invitrogen) and 1% penicillin/streptomycin. Cells were incubated at 37 °C in a humidified atmosphere of 5% CO2. For sphere formation, cells were cultured in serum-free Dulbecco’s modified Eagle’s medium-F12 (Invitrogen), a stem cell-permissive medium containing 20 ng/mL of epidermal growth factor, basic fibroblast growth factor, and B27 (1:50) under sphere-forming conditions. Spheres were collected after 10 days, and the protein was extracted for Western blotting.

### 4.2. Small Interfering RNA (siRNA) Transfection

Cells were transfected at 1 × 105 with 20 nM siRNA targeting HSPA1L (Santa Cruz) or Stealth RNAiTM Negative Control Medium GC (Invitrogen) using Lipofectamine^®^ RNAiMAX reagent (Invitrogen). Cells were incubated for 72 h after transfection, then harvested for RT-PCR and Western blot analyses.

### 4.3. Construction and Transfection of the HSPA1L Overexpression Vector

Poly(A) mRNA was isolated from A549 cells, and a 1925-bp insert of human HSPA1L mRNA was amplified via RT-PCR using the primers BamHI/forward, 5′- ACG GAT CCA TGG CTA CTG CCA AGG GAA TC-3′, and EcoRI/reverse, 5′-ACG AAT TCT TAA TCT ACT TCT TCA ATT GTG GGG-3′. HSPA1L cDNA inserts were cloned into the mammalian expression vector pcDNA3.1 (Invitrogen), and the resultant expression vector (pcHSPA1L) was transfected into cells using Lipofectamine^®^ 2000 (Invitrogen).

### 4.4. Anchorage-Dependent Colony-Formation (Soft Agar Assay) Assays

Cells were plated in 35-mm culture dishes at 1 × 103 cells per plate, incubated for 10 days, and stained with 0.5% crystal violet. Colonies, defined as groups of ≥50 cells, were counted, and the relative colony-forming percentage was plotted, and γ-irradiation was induced by exposure to a single dose of 6 Gy (60Co gamma ray source).

### 4.5. Sphere-Formation Assays

Cells were placed in stem cell-permissive Dulbecco’s Modified Eagle Medium (DMEM-F12; Invitrogen) containing epidermal growth factor (20 ng/mL), basic fibroblast growth factor (20 ng/mL), and B27 Serum-Free Supplement (Invitrogen). Suspended cells were dispensed into ultra-low-attachment 96-well plates (Corning, Inc., Corning, NY, USA) at a density of 1 or 2 cells/well and incubated at 37 °C in a 5% CO_2_ humidified incubator. The next day, each well was visually checked for the presence of a single cell and after 10–14 days, spheres were quantitated using inverted phase contrast microscopy and photographed.

### 4.6. Wound-Healing Assays

Cells were transfected with HSPA1L-specific siRNA or HSPA1L overexpression vectors. After 48 h, the cells were replated in 35-mm culture dishes and cultured until the monolayer was 100% confluent. Cells were then serum-starved overnight by incubation in RPMI 1640 medium containing 0.5% FBS, then a scratch wound was introduced with a plastic pipette tip. The medium was replaced with fresh growth medium, and images were taken at regular intervals over 12–24 h. Cell migration was quantified by measuring the distances between 10 randomly selected points within the wound edge, and the mean values and standard deviations were plotted.

### 4.7. Migration Assays

The lower culture chamber of a 24-transwell plate (Cell Biolabs) was filled with 500 µL migration medium, consisting of RPMI 1640 and 10% FBS. Cells were seeded in the upper chamber at 2 × 105 cells in 200 µL of serum-free medium/well and incubated for 24 h at 37 °C in a humidified atmosphere of 5% CO_2_. Nonmigratory cells in the upper chamber were removed by wiping with a cotton swab. Migratory cells on the bottom of the chambers were stained with crystal violet and counted under a light microscope.

### 4.8. Invasion Assay

Cell invasion was determined using Matrigel-coated invasion chambers (8-μm pores; BD Biosciences, San Jose, CA, USA) per the manufacturer’s instructions. Cells were resuspended in serum-free RPMI 1640 and placed in the upper invasion chamber at 4 × 105 cells/well, then RPMI 1640 containing 10% (*v*/*v*) FBS was added to the lower chamber. The plates were incubated at 37 °C in a humidified atmosphere of 5% CO_2_ for 24 h, and noninvasive cells in the upper chamber were removed by wiping with a cotton swab. Invasive cells on the underside of the inserts were fixed with 4% (*w*/*v*) formaldehyde in phosphate-buffered saline (PBS) and stained with 2% (*w*/*v*) crystal violet in 2% (*v*/*v*) ethanol. Stained cells that had penetrated the Matrigel were counted under a light microscope.

### 4.9. Western Blot Analysis

The anti-TM4SF4 antibody for the Western blot analysis was purchased from Sigma-Aldrich. Antibodies against Sox2, Phospho-AKT (Ser473), AKT, Phospho-NF-κB p65 (Ser536), NF-κB, Phospho-IGF1Rβ (Tyr1131), IGF1Rβ, integrin αV, CD44, and β-actin (Cell Signaling Technology, Danvers, MA, USA); Twist, c-Myc, Cyclin D1, β-catenin (Santa Cruz, Dallas, TX, USA), E-cadherin, and N-cadherin (BD Biosciences, San Jose, CA, USA); Vimentin (Thermo Fisher Scientific, Fremont, CA, USA); and Snail, Notch2, HSPA1L, ALDH1A1, ALDH1A3 (Abcam, Cambridge, UK), and Oct4 (Millipore, Billerica, MA, USA) were used. Protein concentrations were determined using a Lowry kit (Bio-Rad, Hercules, CA, USA). Equal amounts of protein were separated on 8% or 12% sodium dodecyl sulfate-polyacrylamide gels and transferred to a nitrocellulose membrane (Hybond; Amersham Pharmacia). The blots were blocked for 1 h at room temperature with blocking buffer (10% nonfat milk in PBS containing 0.1% Tween 20). The membrane was incubated overnight in a cold chamber with specific antibodies. After being washed with TBS, blots were developed with a peroxidase conjugated secondary antibody, and proteins were visualized via enhanced chemiluminescence procedures (Amersham) following the manufacturer’s protocol.

### 4.10. ALDEFLUOR Assay and FACS

The ALDEFLUOR assay (STEMCELL Technologies, Vancouver, BC, Canada) was performed to isolate and characterize CSC populations in A549 cells per the manufacturer’s instructions. Briefly, 1 × 106 cells were resuspended in an ALDEFLUOR assay buffer containing the ALDH substrate. As a negative control, an aliquot of ALDEFLUOR-exposed cells was immediately quenched with the ALDH inhibitor, diethylaminobenzaldehyde. After incubating for 30 min at 37 °C, the cells were washed and sorted as ALDH+ and ALDH− cells using a FACSAria flow cytometer (BD Biosciences).

### 4.11. Fluorescence Microscopy

Cells (5 × 104 cells/mL) cultured on glass coverslips in six-well plates were fixed with 4% paraformaldehyde and incubated with the primary antibodies in PBS with 1% BSA and 0.1% Triton X-100 at 4 °C overnight. The antibodies used were CD44, β-catenin, E-cadherin, Vimentin, and HSPA1L. Staining was visualized using an Alexa Flour488-conjugated anti-rabbit antibody (Invitrogen). Nuclei were counterstained using 4,6-diamidino-2-phenylindole (DAPI; Sigma-Aldrich, St. Louis, MO, USA). Stained cells were visualized via fluorescence microscopy (Olympus BX53F).

### 4.12. Immunoprecipitation

Cells were lysed in a TX100 lysis buffer (a 2 0 mM Tris-HCl (pH 7.5) buffer containing 150 mM NaCl, 1 mM EGTA, 1 mM EDTA, and 0.5% Triton X-100) and protease inhibitor cocktails (Sigma-Aldrich, St. Louis, MO, USA). Protein concentration was measured using the Bradford reagent (Bio-Rad, Hercules, CA, USA). Immunoprecipitations were performed overnight at 4 °C using 2 mg cell lysate with the appropriate amount of specific antibodies and protein A/G Ultralink Resin (Invitrogen). After washing with TX100 lysis buffer, immunoprecipitates were resuspended in 2× SDS sample buffer, separated using gels, and analyzed by Western blotting using specific antibodies.

### 4.13. Chromatin Immunoprecipitation (ChIP) Assays

ChIP assays were performed using the chromatin immunoprecipitation assay kit from Upstate. Briefly, 1 × 107 cells were treated with 1% formaldehyde at room temperature for 10 min to form cross-links. Cells were washed twice with PBS and lysed in SDS lysis buffer (1% SDS, 10 mM EDTA, 50 mM Tris, pH 8.1, and protease cocktail inhibitor (Roche)) for 10 min on ice. Cells were then sonicated 5 times, for 10 sec each time, with a 1 min rest between rounds to shear the chromatin. Samples were divided into three parts: input, antibody+ (Ab+), and serum (IgG). Ab+ and IgG samples were precleared with salmon sperm DNA/protein agarose-A beads. Ab+ samples were exposed to precipitating antibody (5 μg) overnight. After immunoprecipitation, the immune complexes were collected by adding salmon sperm DNA/protein agarose-A beads. Immune complexes were washed and eluted, and the cross-links reversed by heating. Precipitated DNA was recovered using proteinase K digestion, phenol extraction, and ethanol precipitation. The β-catenin promoter regions were PCR-amplified from the Ab+, IgG, and input DNA samples. Primer sequences for RT-PCR are presented in Table 1. Recovered DNA was amplified using 34 amplification cycles (94 °C for 1 min, 55 °C for 1 min, and 72 °C for 1 min). Assays were repeated three times to confirm reproducibility of the PCR results.

### 4.14. Preparation of Nuclear/Cytoplasmic Fractions

Cells were washed in ice-cold PBS, scraped from the plate, and centrifuged (1000 rpm, 5 min, 4 °C). Cell pellets were resuspended in a lysis buffer (10 mM Tris/HCl, pH 7.4, 10 mM NaCl, 3 mM MgCl2, 0.5% Nonidet P40, and protease inhibitors), then applied on top of 6 mL of a sucrose buffer (0.7 M sucrose, 60 mM KCl, 15 mM NaCl, 15 mM Tris/HCl, pH 7.5, 2 mM EDTA, 0.5 mM EGTA, 14 mM 2-mercaptoethanol, and 0.1% Triton X-100). After centrifuging for 10 min (3500 rpm, 4 °C), the cytoplasmic fraction was harvested from the top of the sucrose buffer, and the nuclei forming a pellet at the bottom of the tube were lysed in RIPA buffer (50 mM Tris/HCl, pH 7.4, 150 mM NaCl, 0.5 mM EDTA, 1% Triton X-100, 0.5% deoxycholate, 0.1% SDS, and 5 μg/mL DNase and protease inhibitors) for 20 min on ice. Cell lysates were then processed as above and centrifuged in the sucrose buffer. Fraction purity was tested via Western blotting for α-tubulin as cytoplasmic markers and HDAC1 as a nuclear marker.

### 4.15. Soft Agar Colony-Formation Assay

Underlayers (2 mL) consisting of 0.9% agar medium were prepared in 60-mm culture dishes by combining 1 volume of 1.8% noble agar (Difco) with 1 volume of 2X RPMI 1640 and 10% FBS. Parental and transfected cells were trypsinized, centrifuged, and resuspended in 0.45% agar medium (1 volume of 0.9% noble agar and 1 volume of 2X RPMI 1640 with 10% FBS). Next, 5 × 105 cells/mL were plated onto the previously prepared underlayers. The cells were incubated at 37 °C in a humidified atmosphere of 5% CO_2_ for 14 days. Colonies were photographed and counted.

### 4.16. CSC Sorting from the A549 NSCLC Cell Line

ALDEFLUORTM assays (STEMCELL Technologies) were performed to isolate and characterize CSC populations from A549 NSCLC cell cultures per the manufacturer’s instructions. Briefly, 106 cells were harvested from cell cultures and resuspended in an ALDEFLUOR assay buffer containing an aldehyde dehydrogenase (ALDH) substrate. As a negative control, an aliquot of ALDEFLUOR-exposed cells was immediately quenched with the specific ALDH inhibitor, N,N-diethylaminobenzaldehyde. After incubating for 30 min at 37 °C, cells were washed and sorted as ALDH1high or ALDH1low cells using a FACSAira flow cytometer (BD Biosciences).

## Figures and Tables

**Figure 1 ijms-21-06957-f001:**
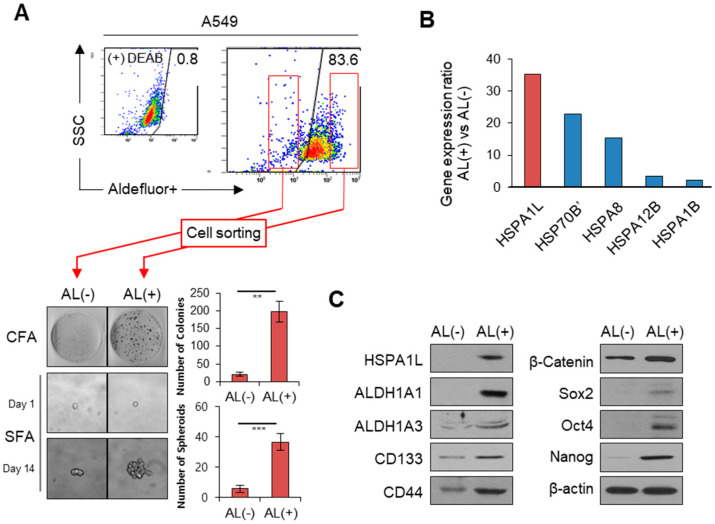
HSPA1L was highly upregulated in ALDH1high cells sorted from A549 cells. (**A**) ADLH1high and ALDH1low subpopulations sorted from A549 cells using ALDEFLUOR. An aliquot of ALDEFLUOR-exposed cells was quenched immediately with the specific ALDH1 inhibitor, diethylaminobenzaldehyde, as the negative control (upper panel). Colony-forming and single-cell sphere-forming assays in ALDH1low (AL−) and ALDH1high (AL+) (lower panel). (**B**) Heat shock protein members differing more than two-fold in gene expression between ALDH1low (AL−) and ALDH1high (AL+). (**C**) Western blot analysis of HSPA1L and CSC markers in ALDH1high and ALDH1low cells. Data represent the mean ± SD of three independent experiments using a two-tailed t-test. ** *p* < 0.005, *** *p* < 0.001.

**Figure 2 ijms-21-06957-f002:**
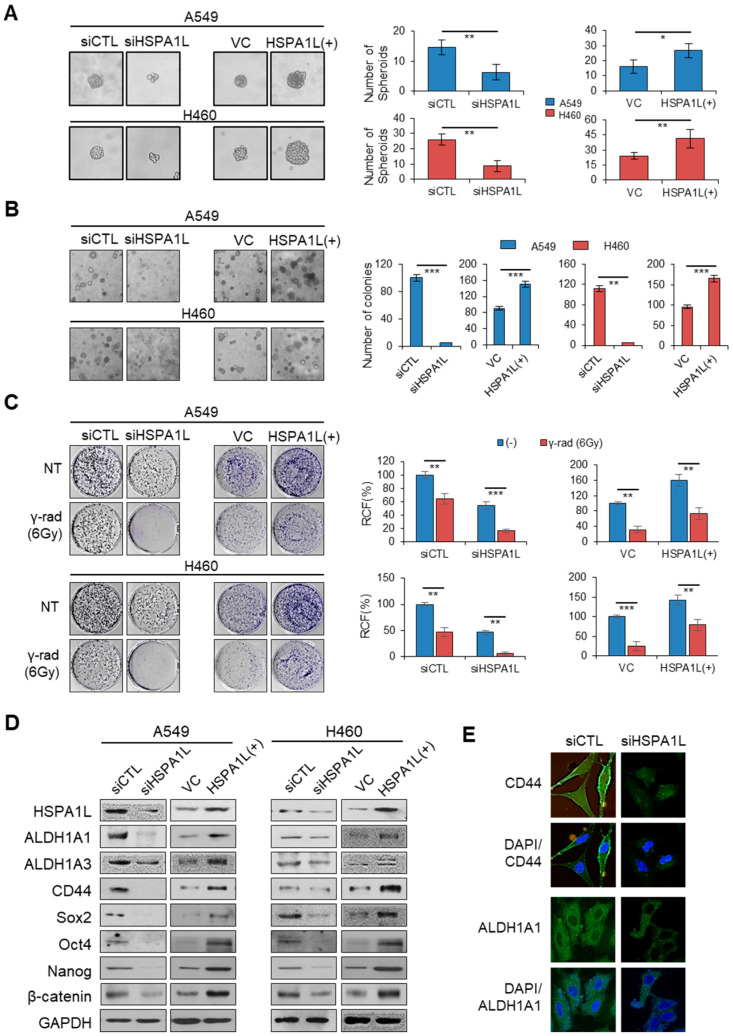
HSPA1L regulates stemness and γ-radiation resistance of lung cancer cells. (**A**) Sphere-forming capacity in A549 and H460 cells transfected with siRNA targeting the HSPA1L and pcDNA-HSPA1L expression vector. (**B**) Anchorage-independent colonization in A549 and H460 cells transfected with siRNA targeting the HSPA1L and pcDNA-HSPA1L expression vector. Cells were photographed under phase-contrast microscopy and quantified. (**C**) Quantification of colony-forming ability in A549 and H460 cells transfected with HSPA1L-targeting siRNA and pcDNA-HSPA1L expression vector; 1 × 103 cells were plated on 35-mm culture dishes 48 h after transfection. Cells were irradiated 24 h later with a single dose of 6 Gy (Dose rate of 0.2 Gy/min). Cells were incubated for 10 days, and colonies were stained with crystal violet and counted, and the relative colony-forming percentage was plotted. (**D**) Western blot analysis of CSC markers ALDH1A1, ALDH1A3, CD44, Sox2, Oct4, Nanog, and β-catenin. GAPDH was used as a loading control. (**E**) Immunocytochemistry analysis of CD44 and ALDH1A1 after transfection with siRNA targeting HSPA1L in A549 cells. Data represent the mean ± SD of three independent experiments using a two-tailed t-test. * *p* < 0.05, ** *p* < 0.01, *** *p* < 0.001.

**Figure 3 ijms-21-06957-f003:**
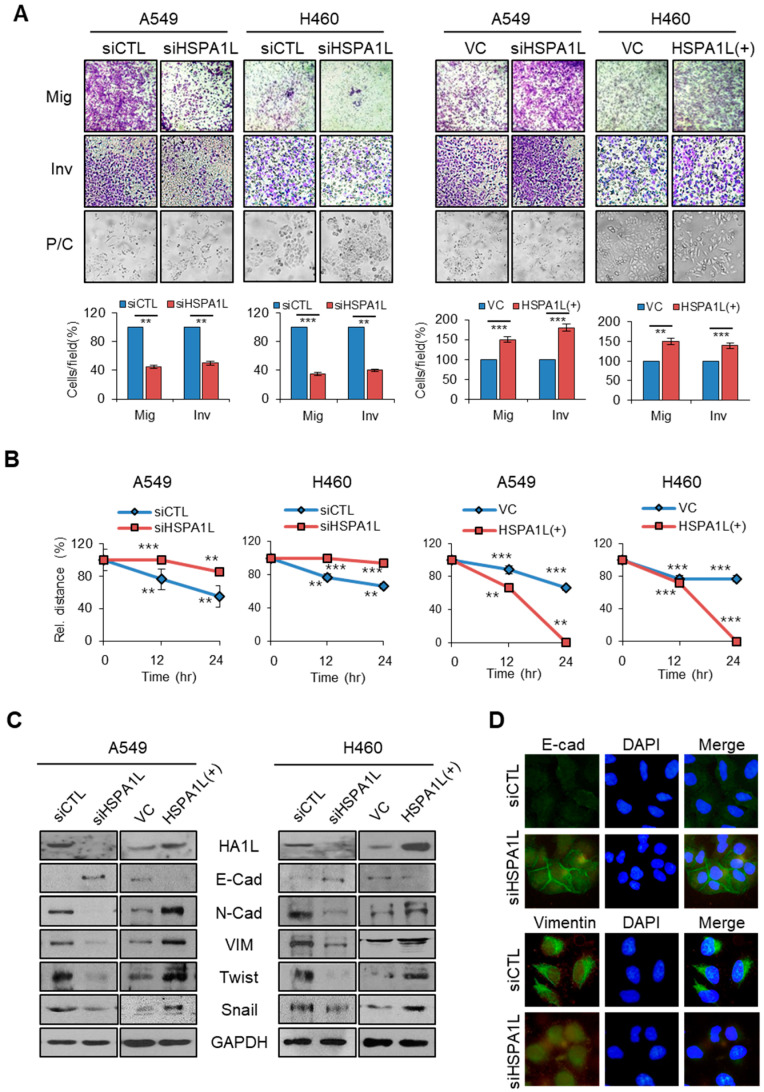
HSPA1L was involved in epithelial–mesenchymal transition (EMT) of lung cancer cells. (**A**) Migration (upper panel) and invasion (middle panel) assays of A549 and H460 cells transfected with the HSPA1L-targeted siRNA and pcDNA-HSPA1L expression vector. Cells in transwells 48 h after transfection. Cells were stained 24 h later and photographed under a phase-contrast (P/C) microscope. The stained cells were counted, and the relative number of cells per field was plotted. (**B**) Wound-healing assays of A549 and H460 lung cancer cells transfected with siRNA targeting the HSPA1L and pcDNA-HSPA1L expression vector. At each point in the assay, cells were photographed, wound size was determined, and the size relative to the initial size was plotted. (**C**) Western blot analysis of EMT markers, E-cadherin, N-cadherin, Vimentin, Twist, and Snail in A549 and H460 cells transfected with siRNA targeting the HSPA1L and pcDNA-HSPA1L expression vector. GAPDH was used as a loading control. (**D**) Immunocytochemistry analysis of E-cadherin and Vimentin after transfection with siRNA targeting the HSPA1L in A549 cells. Target molecules were probed using specific antibodies. Data represent the mean ± SD of three independent experiments using a two-tailed t-test. ** *p* < 0.01, *** *p* < 0.001.

**Figure 4 ijms-21-06957-f004:**
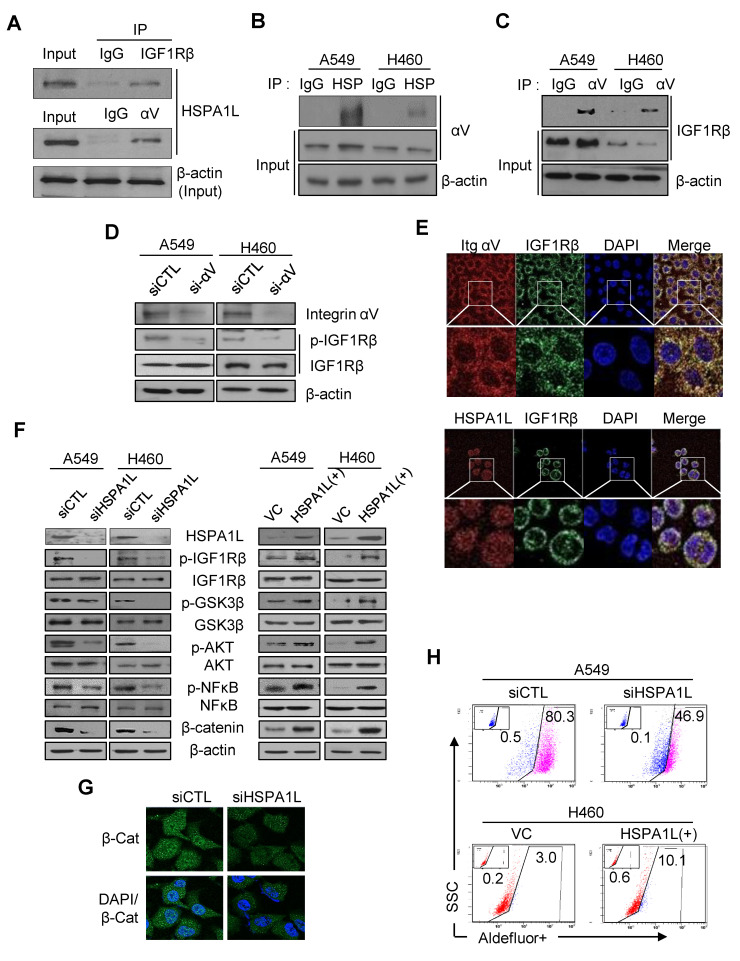
Binding of the HSPA1L/integrin αV complex to IGF1Rβ led to autophosphorylation of IGF1Rβ and activation of the downstream signaling pathways. (**A**) Immunoprecipitation analysis of HSPA1L bound to IGF1Rβ and integrin αV in A549 cells; αV: integrin αV. (**B**) Immunoprecipitation analysis of integrin αV bound to HSPA1L in A549 and H460 cells (**C**) Immunoprecipitation analysis of the interaction between IGF1Rβ and integrin αV in A549 and H460 cells. (**D**) Western blot analysis of p-IGF1Rβ in A549 and H460 cells transfected with siRNA targeting integrin αV. (**E**) Confocal analysis of the interaction between HSPA1L and IGF1Rβ in H460 cells or integrin αV and IGF1Rβ in A549 cells. (**F**) Western blot analysis of the activation status of the IGF1Rβ/AKT/GSK3β or AKT/NFkB pathway after transfection with siRNA targeting HSPA1L (left panel) and with the pcDNA-HSPA1L expression vector (right panel). (**G**) Immunocytochemistry analysis of β-catenin after transfection with siRNA targeting HSPA1L in A549 cells. (**H**) FACS analysis of transfection with siRNA targeting the HSPA1L and pcDNA-HSPA1L expression vector in A549 and H460 lung cancer cells.

**Figure 5 ijms-21-06957-f005:**
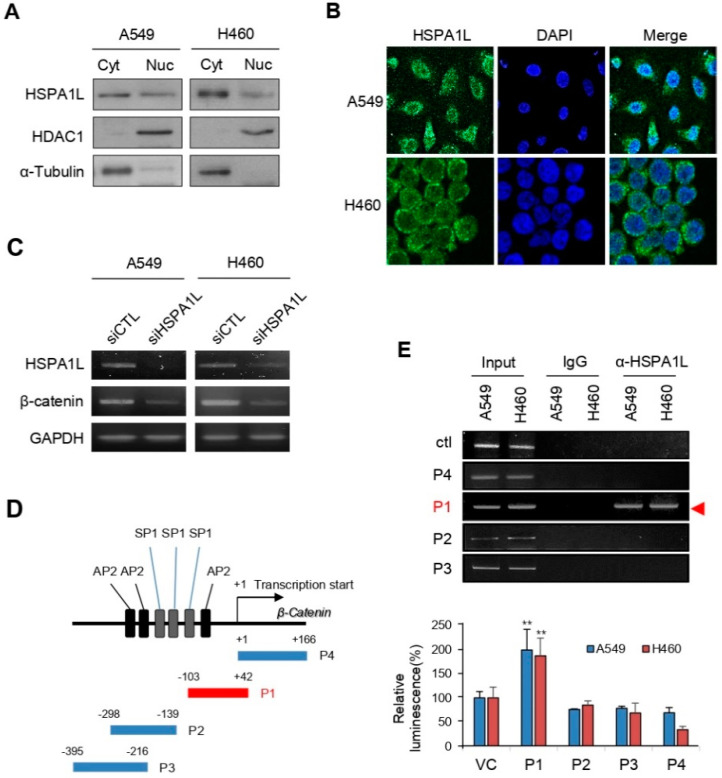
HSPA1L directly binds to a specific β-catenin promoter region and regulates β-catenin gene expression. (**A**) Different subcellular fractions of A549 and H460 cells were prepared; Cyt: cytosolic fraction; Nuc: nuclear fraction. Western blot analysis indicated that HSPA1L was localized in both the cytoplasm and the nucleus. (**B**) Immunocytochemistry analysis of HSPA1L localization; HSPA1L localized in both the cytoplasm and the nucleus in A549 and H460 cells. (**C**) HSPA1L regulated β-catenin gene expression. (**D**) Nucleotide sequence of the 5′-flanking region and noncoding exon 1. Schematic diagrams of the potential promoter regions of the β-catenin gene. (**E**) ChIP analysis of HSPA1L binding to potential promoter regions of β-catenin in A549 and H460 cells. RT-PCR was performed using primers specific to the promoter region. Luciferase activities were measured after transfecting the promoter construct in A549 and H460 cells. Data represent the mean ± SD of three independent experiments using a two-tailed t-test. ** *p* < 0.01.

**Figure 6 ijms-21-06957-f006:**
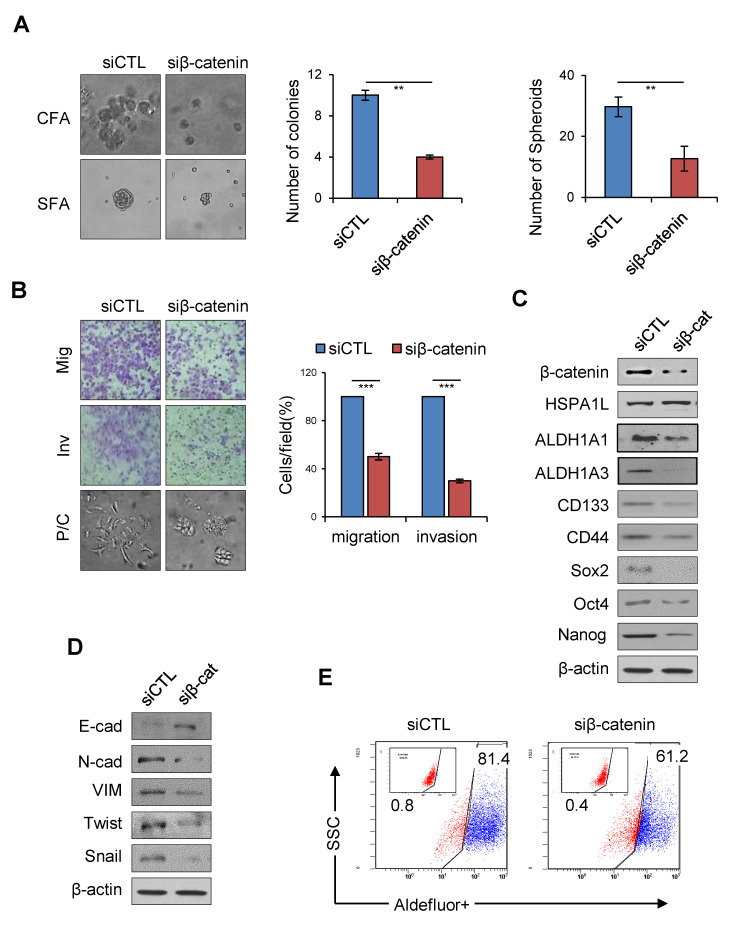
HSPA1L/β-catenin axis was involved in stemness by regulating ALDH1 expression. (**A**) Anchorage-independent colony and sphere formation of lung cancer in A549 cells transfected with siRNA targeting β-catenin. (**B**) Migration (upper panel), invasion (middle panel), and morphological (lower panel) assay in A549 cells transfected with siRNA targeting β-catenin. Cells were stained 72 h after transfection and photographed under phase-contrast microscopy. The stained cells were counted, and the relative number of cells per field was plotted. (**C**) Western blot analysis for CSC markers ALDH1A1, ALDH1A3, CD133, CD44, Sox2, Oct4, and Nanog in A549 cells transfected with siRNA targeting β-catenin. (**D**) Western blot analysis of EMT markers, E-cadherin, N-cadherin, Vimentin, Twist, and Snail in A549 cells transfected with siRNA targeting β-catenin. β-actin was used as a loading control. (**E**) FACS analysis in A549 cells transfected with siRNA targeting β-catenin. Data represent the mean ±SD of three independent experiments using a two-tailed t-test. ** *p* < 0.01, *** *p* < 0.001.

**Figure 7 ijms-21-06957-f007:**
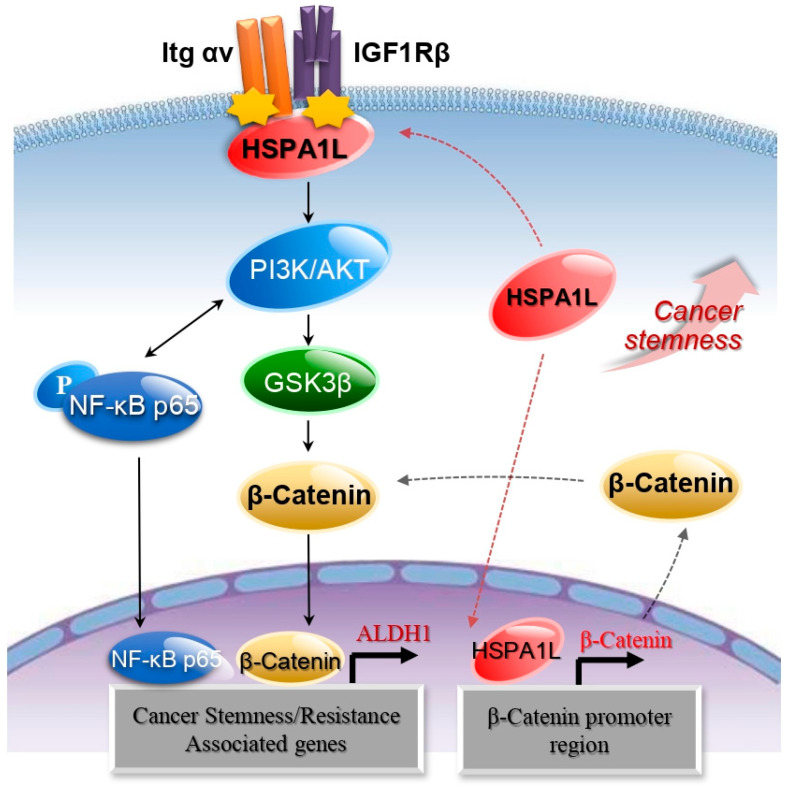
Schematic model for two-way functions of HSPA1L between the cytosol and nucleus. HSPA1L promoted EMT and tumorigenic capacity through β-catenin expression.

**Table 1 ijms-21-06957-t001:** Promoter primer sequences of *β-catenin* in different promoter sites using the HSPA1L antibody.

Primer	Sequence
P1-fw	GCCCCTTGTCCTCGCGCGGCGGAA
P1-rv	CCCGGGGCCGGGCCAACGCTGCTG
P2-fw	GGGGGCCCGGCCTCCCCGATGCAG
P2-rv	CCGCCCAGCAGTCTGCTGTGACGG
P3-fw	CTGCAGCTGCTCTCCCGG
P3-rv	GTGACGGCTGGCGGCTGC
P4-fw	AAGCCTCTCGGTCTGTGG
P4-rv	CTCAGGGGAACAGGCTCC

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
