# Peer review of "HSPA1L Enhances Cancer Stem Cell-Like Properties by Activating IGF1Rβ and Regulating β-Catenin Transcription"

_ijms, 2020, doi:10.3390/ijms21186957_

Round 1

Reviewer 1 Report

The manuscript by Choi et al. reveals that HSPA1L protein induced by ALDH1, known as a CSCs marker, promotes self-renewal and tumorigenic capacity by activating AKT/NF-κB p65 and AKT/GSK3β/β-catenin pathway in lung cancer. In this manuscript, overall results are interesting and clearly presented, but the relationship between ALDH1 and HSPA1L seems to be further elucidated and there are some points that should be carefully revised as described below.

The minor concerns on this study were described as below:

  1. As a cancer stem cell marker, it seems necessary to explain the reason for focusing on ALDH1 among various markers other than transmembrane glycoproteins, CD44, CD133, and CD326, which have been suggested in the paper.
  2. In figure 1A, it seems that the sphere formation is small even in low ALDH1 expressed cells, which is indicated as a major marker of CSCs. It is necessary to further explain how critical ALDH1 is as a CSCs marker in lung cancer cells.
  3. In Figure 1B, in DNA microarray analysis, the expression of HSPA1L gene expression differs by about 35 times according to the ALDH1 (+) vs. ALDH1 (-) ratio. In this system, validation of the mRNA of this gene is required.
  4. In the sphere formation assay and colony formation assay (Figure 1 A), it seems necessary to express clearly by quantifying the number of colonies of the size shown in the picture in a graph.
  5. It is difficult to confirm the ability of anchorage-independent colonization by HSPA1L because the photo resolution in Figure 2B is low and there is no difference in colony size between groups.
  6. As a result of western blotting after transfection of siHSPA1L into H460 cells in Figure 2D, there appears to be no difference in expression of ALDH1A1, ALDH1A3, and CD44. In addition, as a result of western blotting after overexpression of ALDH1A, the expression of ALDH1A3 was faint and no difference was known.
  7. In Figure 2E, the labeling of siHSPA1L seems to need to be modified.
  8. In Figure 3C, the degree of ALDH1A overexpression in A549 cells is too weak, and the expression of Snail, an EMT marker, is decreased. The results of the text show that it is increased, so the experimental results and explanations do not agree. In addition, the western blotting results after siHSPA1L transfection into H460 cells show an increasing pattern of snail expression, which does not match the contents of the result.
  9. In the scheme shown in Figure 7, additional experiments are needed to see if HSPA1L can affect β-catenin and phosphorylated NF-kB, acting as a transcription factor and affecting the expression of ALDH1.

Author Response

1. As a cancer stem cell marker, it seems necessary to explain the reason for focusing on ALDH1 among various markers other than transmembrane glycoproteins, CD44, CD133, and CD326, which have been suggested in the paper.

Response: Since ALDH1 is involved in self-renewal, differentiation and self-protection, its importance as a biomarker of cancer stem cells or normal stem cells has been recognized. For this reason, we focused our research on ALDH1 as an important factor in determining cancer stem cell properties. A sentence was inserted in the introduction (blue color).

2. In figure 1A, it seems that the sphere formation is small even in low ALDH1 expressed cells, which is indicated as a major marker of CSCs. It is necessary to further explain how critical ALDH1 is as a CSCs marker in lung cancer cells.

Response: Aldehyde dehydrogenase 1(ALDH1) has been suggested an important biological marker of CSCs in lung cancer cell lines. Therefore, since data related to changes in self-renewal capacity following ALDH1 suppression or overexpression have been presented in many papers including ours, this data was not presented in this study. Please refer to the pictures.

3. In Figure 1B, in DNA microarray analysis, the expression of HSPA1L gene expression differs by about 35 times according to the ALDH1 (+) vs. ALDH1 (-) ratio. In this system, validation of the mRNA of this gene is required.

Response: As pointed out by the reviewer, it is appropriate to additionally confirm the mRNA levels of each isoform by RT-PCR analysis. However, it is difficult to experiment due to the time limit of the response. Instead, raw data from DNA chip analysis were presented to the supplement.

4. In the sphere formation assay and colony formation assay (Figure 1 A), it seems necessary to express clearly by quantifying the number of colonies of the size shown in the picture in a graph.

Response: According to reviewer’s comments, statistical data on colony formation and sphere formation were presented.

5. It is difficult to confirm the ability of anchorage-independent colonization by HSPA1L because the photo resolution in Figure 2B is low and there is no difference in colony size between groups.

Response: The presented photos are reduced, so the difference in number or size seems to be unclear, but the actual original photos clearly show the difference.

6. As a result of western blotting after transfection of siHSPA1L into H460 cells in Figure 2D, there appears to be no difference in expression of ALDH1A1, ALDH1A3, and CD44. In addition, as a result of western blotting after overexpression of ALDH1A, the expression of ALDH1A3 was faint and no difference was known.

Response: H460 cells are NSCLC cell lines with little ALDH1 (below 1-2%). The H460 cell line with low ALDH1 was selected and used because it seemed to be clear to see the effect of HSPA1L overexpression. However, even if HSPA1L is overexpressed, the FACS data by ALDOFLUOR does not exceed 10-15%. Therefore, despite the ambiguity, there is a significant difference as a result of confirming with Western blot and Anchorage independent colonization.

7. In Figure 2E, the labeling of siHSPA1L seems to need to be modified.

Response: According to reviewer’s comments, we corrected the labeling of siHSPA1L appropriately.

8. In Figure 3C, the degree of ALDH1A overexpression in A549 cells is too weak, and the expression of Snail, an EMT marker, is decreased. The results of the text show that it is increased, so the experimental results and explanations do not agree. In addition, the western blotting results after siHSPA1L transfection into H460 cells show an increasing pattern of snail expression, which does not match the contents of the result.

Response: The reviewer's point is correct. In the process of moving the picture, the A549 HSPA1L(+) picture and the H460(siHSP1AL) picture were changed.

9. In the scheme shown in Figure 7, additional experiments are needed to see if HSPA1L can affect β-catenin and phosphorylated NF-kB, acting as a transcription factor and affecting the expression of ALDH1.

Response: In preparation for a new paper, studies are underway on genes that the HSPA1L (or other proteins)/NFkB-p65 axis regulates and may affect cancer stemness.

Reviewer 2 Report

In this paper, the authors showed how HSPA1L can modulate EMT-related events.

The authors explored two roles of HSPA1L, one as a transcription factor and the other one as a receptor regulator, using many technical approaches. Even if the transcriptional role of HSPA1L might have been strengthened, this work remains important in the field.

Major points:

  • In fig1B the authors showed what they called DNA microarray results. Isn't it RNA microarray results. This point should be clarified, especially thanks to a paragraph in material and methods. Moreover, some journals ask for the public availability of such raw data.
  • Fig2 legend and corresponding results should be harmonized as the techniques described in both parts seem not correspond each other, sometimes. Moreover single-cell forming material and methods is missing.
  • In fig2D, the protein levels of many targets is different between siCTL and VC conditions. Even if cells were subjected to different transfection conditions, can it explain the huge observed differences?
  • The authors must search for the impact of HSPA1L down and upregulation effects on cell death. An impact of transfection (controls), overexpression or downregulation of HSPA1L on cell viability might impact results on migration, invasion sphere formation, ....  Please look for cell death features (membrane permeabilization at least, and annexinV staining).
  • In fig3C, Snail expression is not consistent.
  • In fig4A to C, why the authors do not search for immunoprecipitated protein expression (A: IGF1R, B: HSPA1L, C: aV)? The material and methods for IP is missing.
  • The discussion must be improved/completed. How HSPA1L could bind on DNA or more certainly on a transcription factor? How to inhibit HSPA1L in therapy and why is it a better way to impact cancer cells than targeting IFG1R?

Minor points:

  • English must be edited.
  • Please temper the deductions on fig4A to C. Line 186 replace "indicating" by suggesting as an interaction cannot certify a modification of the activation. Lane 190 remove "direct". A direct interactoin should be shown using recombinant proteins as cellular IP cannot exclude intermediate partners. Lane 193, confocal cannot show interaction, but only colocalizations. The same, Lane 329 change "bound directly".
  • red stainings in fig4 are not optimal.
  • Lane 272: the fact that HSPA1L was not affected by sib-catenin should be precised as it show that HSPA1L is upstream.

Author Response

Major points:

  • In fig1B the authors showed what they called DNA microarray results. Isn't it RNA microarray results. This point should be clarified, especially thanks to a paragraph in material and methods. Moreover, some journals ask for the public availability of such raw data.

Response : Our results were obtained by mRNA expression cDNA microarray. To be clearer, we replaced the term “DNA microarray” to “cDNA microarray” in the manuscript. The raw data of microarray are not publically available now. If the journal requires the raw data, we can provided it in the supplementary data.

  • Fig2 legend and corresponding results should be harmonized as the techniques described in both parts seem not correspond each other, sometimes. Moreover single-cell forming material and methods is missing.

Response : As you pointed out, we revised the manuscript so that the description on the methods to be matched in both figure legends and results. (Ex. Anchorage-dependent colonization (agarose gel assay)). In addition, the methods of sphere-formation assay was added to Material and Methods as follows.

Sphere-formation assay

Cells were placed in stem cell–permissive Dulbecco’s Modified Eagle Medium (DMEM-F12; Invitrogen) containing epidermal growth factor(20 ng/ml), basic fibroblast growth factor(20 ng/ml), and B27 Serum-Free Supplement (Invitrogen). Suspended cells were dispensed into ultra-low-attachment 96-well plates (Corning, Inc., Corning, NY, USA) at a density of 1 or 2 cells/well and incubated at 37°C in a 5% CO2 humidified incubator. The next day, each well was visually checked for the presence of a single cell and after 10–14 days, spheres were quantitated using inverted phase contrast microscopy and photographed.

  • In fig2D, the protein levels of many targets is different between siCTL and VC conditions. Even if cells were subjected to different transfection conditions, can it explain the huge observed differences?

Response: Since the transfection methods are different for two methods, the siRNA knockdown experiment and the overexpression vector experiment were not performed simultaneously. Additionally, to obtain the image showing the difference between other groups more prominently, different exposure times were applied during Western blot development, and this may have resulted the different intensities between siRNA and overexpression experiments.

  • The authors must search for the impact of HSPA1L down and upregulation effects on cell death. An impact of transfection (controls), overexpression or downregulation of HSPA1L on cell viability might impact results on migration, invasion sphere formation,  Please look for cell death features (membrane permeabilization at least, and annexinV staining).

Response: As you pointed out, transfection process affects the cell viability and growth. However, we assume that this effect can be offset by including siCTL group as a control group. We did not directly examine the effect of HSPA1L expression on cell viability or proliferation. However, as shown in the figure attached below (photographical result of Fig. 3B), we could not detect noticeable cell death by overexpression and inhibition of the HSPA1L gene. Also we think that these properties are also the aspects of cancer stemness thus the effects on the cell death and proliferation are already reflected in the results of the migration, invasion, and sphere formation assays.

  • In fig3C, Snail expression is not consistent.
    Response: The reviewer's point is correct. In the process of editing the picture, there was an error that the photo of A549 HSPA1L(+) was exchanged with that of H460(siHSP1AL). We corrected this error..

  • In fig4A to C, why the authors do not search for immunoprecipitated protein expression (A: IGF1R, B: HSPA1L, C: aV)? The material and methods for IP is missing.

Response: There are many ways to express immunoprecipitation experiment results. It is judged that the input result can complement the immunoprecipitated protein expression (A: IGF1R, B: HSPA1L, C: aV) results. In addition, information about Immunoprecipitation is added to materials and methods as follows.

Immunoprecipitation

Cells were lysed in TX100 lysis buffer[20 mM Tris-HCl (pH 7.5) buffer containing 150 mM NaCl, 1 mM EGTA, 1 mM EDTA, 0.5% Triton X-100] and protease inhibitor cocktails (Sigma-Aldrich, St Louis, MO, USA). Protein concentration was measured using the Bradford reagent (Bio-Rad, Hercules, CA, USA). Immunoprecipitations were performed overnight at 4°C using 2 mg cell lysate with the appropriate amount of specific antibodies and protein A/G Ultralink Resin (Invitrogen). After washing with TX100 lysis buffer, immunoprecipitates were resuspended in 2× SDS sample buffer, separated using gels, and analysed by Western blotting using specific antibodies.

  • The discussion must be improved/completed. How HSPA1L could bind on DNA or more certainly on a transcription factor? How to inhibit HSPA1L in therapy and why is it a better way to impact cancer cells than targeting IFG1R?

Response: Further study will be conducted to clarify how HSPA1L interacts with DNA or transcription factor. Unfortunately, we cannot give the answer right now because of the revision due date. IGF1R accepts a variety of signals and, in particular, and it is a very important receptor for insulin-related signaling. So it cannot be used as a direct target because a lot of side effects can be caused when targeting the IGF1R. On the other hand, HSPA1L seems to be a cancer-specific target downstream of IGF1R. Therefore targeting HSPA1L can sufficiently suppress IGF1R signaling but cause less side-effects than direct IGF-1R targeting.

Minor points:

  • English must be edited.

Response: This manuscript has already been edited by a proofreading expert.

  • Please temper the deductions on fig4A to C. Line 186 replace "indicating" by suggesting as an interaction cannot certify a modification of the activation. Lane 190 remove "direct". A direct interactoin should be shown using recombinant proteins as cellular IP cannot exclude intermediate partners. Lane 193, confocal cannot show interaction, but only colocalizations. The same, Lane 329 change "bound directly".
    Response: We revised the sentence as you suggested.

  • red stainings in fig4 are not optimal.

Response: Unfortunately, this is the best picture currently we have. There is not enough time to obtain new photo with optimal staining condition before the revision due time.

  • Lane 272: the fact that HSPA1L was not affected by sib-catenin should be precised as it show that HSPA1L is upstream.

Response: As you pointed out, our data in Fig. 6 showed that si-beta-catenin does not affect the HSPA1L supporting that HSPA1L is an upstream regulator of beta-catenin. The result description was added in Result section.

Reviewer 3 Report

The manuscript describes an important role of HSPA1L in CSC, especially lung cancers. Specifically, they demonstrate that both of cytoplasmic and nuclueic HSPA1Ls and Wnt/b-catenin pathway are responsible for cancer stem cell like properties in lung cancer cells. This has important implications for future drug or biomarker design, but the authors need to address the following concerns to establish how applicable these findings are to a broader range of lung cancer patients.     

Concerns:

* Why did the authors select lung cancer to verify the role of HSPA1L? Please include a more detail rationale for choosing NSCLC for studying HSPA1L and ALDH1 (or correIation with GF1R pathway) in this study.

* Why were other NSCLC cells, such as H292, H838, or H1650, not used in this study? Please include the rationale (eg. The levels of HSPA1L or corrleation with ALDH1) for testing A549 and H460 to further emphasize on the importance of HSPA1L in NSCLC.

* Could the authors include the ChIP data for checking the direct binding HSPA1L via DNA binding domain? It would be further support the direct necleus role of HSPA1L in figure 5E. Furthermore, please check whether HSPA1L could regulate Wnt/b-catenin pathway under AL (+) conditions in experiments of Figure 1 to verify the importance of AL(+)-HSP1AL-b-cetnin axis.

* Please check localization of b-catenin under presence or absence of HSPA1L using fractionation or immunofluorescence, if possible.

* How frequent are HSP1AL alterations in NSCLC? (mutations, amplifications, deletions, mRNA/protein over/under-expression, please use public database, if possible)

Author Response

Responses to Reviewer 3.

* Why were other NSCLC cells, such as H292, H838, or H1650, not used in this study? Please include the rationale (eg. The levels of HSPA1L or corrleation with ALDH1) for testing A549 and H460 to further emphasize on the importance of HSPA1L in NSCLC.

Response: Among lung cancer, NSCLC is very difficult to treat because of its resistance to radiation and drugs, which are typical characteristics of cancer stem cells. Therefore, to investigate whether HSPA1L was involved in the enrichment of stem cell in lung cancer cells, this study was performed using A549, an adenocarcinoma cell line with a lot of ALDH1 and high radiation resistance, and H460 cells with a relatively low ALDH1 and low radiation resistance. A sentence was inserted in the introduction(blue color)

* Why did the authors select lung cancer to verify the role of HSPA1L? Please include a more detail rationale for choosing NSCLC for studying HSPA1L and ALDH1 (or correIation with IGF1R pathway) in this study.

Response: We are doing research on lung cancer cells that are resistant to radiation therapy. To find out the cause of this, an experiment was conducted using a malignant and radiation resistant lung cancer cell line (NSCLC cells) among lung cancer cells, and the experiment was conducted by confirming that the expression of HSPA1L was high in CSC-like ALDH1high cell with radiation resistance ability.

* Could the authors include the ChIP data for checking the direct binding HSPA1L via DNA binding domain? It would be further support the direct necleus role of HSPA1L in figure 5E. Furthermore, please check whether HSPA1L could regulate Wnt/b-catenin pathway under AL (+) conditions in experiments of Figure 1 to verify the importance of AL(+)-HSP1AL-b-cetnin axis.

Response: The experiment to find the exact DNA binding domain to which HSPA1L binds will be conducted as a future study. In addition, we plan to further study Notch and SHH signaling mechanisms involved in CSCs, as well as Wnt-β-catenin signaling mechanisms.

* Please check localization of b-catenin under presence or absence of HSPA1L using fractionation or immunofluorescence, if possible.

Response: We performed immunofluorescence of b-catenin with HSPA1L suppression (Fig 4G).    b-catenin is distributed in both cytosol and nucleus

* How frequent are HSP1AL alterations in NSCLC? (mutations, amplifications, deletions, mRNA/protein over/under-expression, please use public database, if possible)

Response: In the case of HSPA1L, there are reports of genetic polymorphism, but no mutations such as deletion have been found. Recently a new rare mutation related to inflammation was reported. However, there are not many studies related to HSPA1L so far, so there are not many reports related to mutation.
